# Cardiac Output Evaluation on Septic Shock Patients: Comparison between Calibrated and Uncalibrated Devices during Vasopressor Therapy

**DOI:** 10.3390/jcm10020213

**Published:** 2021-01-09

**Authors:** Paolo Persona, Ilaria Valeri, Elisabetta Saraceni, Alessandro De Cassai, Fabrizia Calabrese, Paolo Navalesi

**Affiliations:** 1UOC Institute of Anaesthesia and Intensive Care Unit, University Hospital of Padua, 35121 Padua, Italy; alessandro.decassai@gmail.com (A.D.C.); fabrycal@hotmail.com (F.C.); 2UOC Institute of Anaesthesia and Intensive Care Unit, Department of Medicine-DIMED, University of Padua, 35121 Padua, Italy; ilaria.valeri.90@gmail.com (I.V.); pnavalesi@gmail.com (P.N.); 3Clinical Department of Anaesthesiology and Intensive Care Medicine, SS. Annunziata Hospital, 66100 Chieti, Italy; betta.saraceni@gmail.com

**Keywords:** hemodynamic monitoring, septic shock, MostCare, transpulmonary thermodilution

## Abstract

There are no reliable, non-invasive methods to accurately measure cardiac output (CO) in septic patients. MostCare (Vytech Health™, Vygon, Padova, Italy), is a beat-to-beat, self calibrated method for CO measurement based on continuous analysis of reflected arterial pressure waveforms. We enrolled 40 patients that were suffering from septic shock and requiring norepinephrine infusion to target blood pressure in order to to evaluate the level of agreement between a calibrated transpulmonary thermodilution device (PiCCO System, Pulsion Medical Systems, Feldkirchen, Germany) and the MostCare system in detecting and tracking changes in CO measurements related to norepinephrine reduction in septic shock patients,. PiCCO was connected to a 5 Fr femoral artery catheter and to a central venous catheter. System calibration was performed with 15 mL of cold saline injection over about 3 s. The MostCare device was connected to the artery catheter to analyze the arterial waveform. Before reducing norepinephrine infusion, the PiCCO system was calibrated, the MostCare waveform was optimized, and the values of the complete hemodynamic profile were recorded (T1). Norepinephrine infusion was then reduced by 0.03 mcg/Kg/min. After 30 min, a new calibration of PiCCO system and a new record on both monitors were performed (T2). Static measurements agreements were assessed using the Bland-Altman test, while trending ability was investigated using polar plot analysis. If volume expansion occurred, then related data were separately analyzed. At T1 mean the CO was 5.38 (SD 0.60) L/min, the mean difference was 0.176 L/min, the limits of agreement (LoA) was +1.39 and −1.04 L/min, and the percentage error (PE) was 22.6%; at T2 the mean CO was 5.44 (SD 0.73) L/min, the mean difference was 0.053 L/min, the LoA was +1.51 and −1.40, and the PE was 27%. After considering the volume expansion between T1 and T2, the mean CO at T1 was 5.39 L/min (SD 0.47), the LoA was +1.09 and −0.78 L/min, and the percentage error (PE) was 17%; at T2 the mean CO was 5.35 L/min (SD 0.81), the LoA was +1.73 and −1.52 L/min, and the PE was 30%. The polar plot diagram seems to confirm the trending ability of MostCare system versus the reference method. In septic patients, when the arterial waveform is accurate, MostCare and PiCCO transpulmonary thermodilution exhibit good agreement even after the reduction of norepinephrine and changes in vascular tone or volume expansion. MostCare could be a rapid to set, reliable, and useful tool to monitor hemodynamic variations in septic patients in emergency contexts where thermodilution methods or other advanced systems are not easily available.

## 1. Introduction

There are no reliable, non-invasive, easy, and rapid setup methods to accurately measure cardiac output (CO) in septic patients in an emergency setting. The invasiveness and the potentially negative impact on outcome [1,2,3] of the reference method of CO assessment, the pulmonary artery catheter (PAC), has encouraged the development of new devices. Nevertheless, some concerns have been raised about the accuracy and precision of CO measurements being performed by different hemodynamic monitoring systems.

As CO, together with the arterial impedance and heart rate, is the main determinant of arterial pressure, many algorithms have been developed in order to calculate CO based on pulse contour analysis of the arterial pressure waveform [4].

Many pulse contour methods are now available for clinical purposes, and most of them need to be calibrated by using an indicator (ice cold saline or lithium); others analyze the geometric variations of the arterial waveform in comparison to an internal database [5,6].

However, especially in septic patients, the main issue of pulse contour methods concerns the lack of measurement of the arterial system impedance [7]. Arterial impedance and elastance in fact often change in sepsis, due to the extreme vasodilation and the extensive use of norepinephrine and vasopressin to maintain vascular tone and mean pressure. For this reason, the re-calibration of transpulmonary thermodilution based devices is considered to be essential to get reliable data on cardiac performance after vasopressor modification [8,9,10,11].

If properly managed, transpulmonary thermodilution methods also appear to not be inferior to PAC for also measuring CO in septic patients [12,13,14].

MostCare (Vytech Health™, Vygon, Padova, Italy) is a beat-to-beat, self-calibrated method for CO measurement, based on continuous analysis of reflected waves recorded on pulse contour at 1000 points/s [15]. Based on Pressure Recording Analytical Method (PRAM), it identifies the points of instability distributed along the systolic and diastolic phases of the arterial waveform; calculations take into account the forward (from the heart) and the backward (from peripheral vessels) waves. The continuous evaluation of cardiac output (CCO) may give a real time assessment of heart performance related to modifications of arterial preload and afterload.

MostCare’s performance on the septic population has not been clarified yet, due to the discordant results of the few available studies [16,17].

The aim of our work was to evaluate the level of agreement between a calibrated transpulmonary thermodilution device (PiCCO System, Pulsion Medical Systems, Feldkirchen, Germany), and the MostCare system in detecting and tracking changes in CCO measurements related to norepinephrine reduction in septic shock patients.

## 2. Methods

We performed a single center observational clinical study.

After local ethic committee approval (Ethical Board protocol n°26658), we considered 50 consecutive patients admitted to our 19 bed ICU for enrollment with the following inclusion criteria: (1) diagnosis of septic shock; (2) norepinephrine infusion; (3) a previous clinical decision to use the PiCCO system for hemodynamic monitoring; (4) age >18 years. Exclusion criteria were: (1) any clinical condition that could interfere with the reliability of the reference method (valvular regurgitations, post-aneurismatic endoleaks, lung resections, etc.) [18]; (2) over or under damped arterial waveform artefacts despite the use of internal or external filters (the MostCare monitor erases some harmonica as part of arterial waveforms that present a too steep ascending line in the systolic phase).

Written informed consent or delayed consent was obtained from each patient or legal surrogate according to the Local Ethics Committee dispositions. This manuscript adheres to the applicable EQUATOR guidelines.

Septic shock was diagnosed according to the criteria reported by Rhodes et al. [19].

Before inclusion in the study, a trained staff physician performed a transthoracic echocardiography; if major valvular defects, intracardiac shunts, or other conditions that could afflict the accuracy of transpulmonary thermodilution method were diagnosed, then the patient was excluded from the study [18]. PiCCO was connected to a 5 Fr femoral artery catheter (Pulsiocath PV2015L20, Pulsion Medical Systems), and to a central venous catheter with the tip at the cavoatrial junction. System calibration was performed with 15 mL of iced saline injection (temperature < 8 °C) for about 3 s. The reproducibility of the measurement was tested by performing 3 subsequent calibrations. If a difference greater than 20% between each CO-thermodilution measurement occurred, then a new calibration series was performed and considered.

The MostCare device was connected to the patient monitor to analyze the arterial waveform. MostCare estimates stroke volume (SV) based on the formula SV = A/Z, where A is the systolic part of the curve and Z is the impedance of the system; Z is calculated as P/t × K, where P is instant pressure and t is time, while considering a beat to beat analysis of the whole cardiac cycle and the reflected waves from the arterial system. The reflected waves cause instability points with different velocities on the pulse contour detected by analysis at 1000 points/sec by MostCare. K is calculated as the ratio between expected and measured mean blood pressures, assuming that the numerator is constant according to Guyton studies [20,21].

The fast-flush test was performed and, if arterial resonance occurred, the signal was optimized with specific devices [22] or using MostCare filters [23]. If the signal, despite the efforts to clear it, was over or under damped based on the already described technique [24], then the patient was excluded from the study. A mean of 10 s consecutive MostCare CCO values were considered for comparison.

Before any clinical decision to reduce norepinephrine, in accordance with our internal protocol, the PiCCO system was calibrated [25], the MostCare waveforms were optimized, and the values of the complete hemodynamic profile were recorded (T1). Norepinephrine infusion was then reduced by 0.03 mcg/Kg/min. After 30 min, a new calibration of PiCCO system and a new record on both monitors were performed (T2). Data from PiCCO and MostCare during the norepinephrine de-escalation were recorded and stored for off-line analysis.

Between T1 and T2 measurements, clinicians were free to perform volume expansion with 500 mL of crystalloids to maintain mean arterial pressure (MAP).

### Statistical Analysis

We planned a study of a continuous response variable using matched pairs of study values. Prior data indicate that the difference in the response of matched pairs is normally distributed with a standard deviation of 1. If the true difference in the mean response of matched pairs was 0.55, then we needed to study 37 subjects (paired evaluations) to be able to reject the null hypothesis that this response difference is zero with a probability (power) of 0.9. The Type I error probability associated with this test of this null hypothesis is 0.05.

To assess whether the values were normally distributed, we performed the Shapiro-Wilk test. Correlations were found using the Pearson product moment coefficient. Static measurements agreements were assessed using the Bland-Altman test [26], considering the limits of agreement (LoA), and the percentage error (PE), to test the precision of the measurement. PE was calculated as the ratio between limits of agreement of the bias and the mean CO of the two systems. According to the literature, a PE of less than 30% is considered to be acceptable for clinical purposes [27]. Trending ability was investigated using polar plot analysis. The polar plot analysis is widely described elsewhere [28]; briefly, the agreement is shown by angle θ made by ΔCO vector with the line of identity (y = x) against the change in the CO as the radian (distance of data point from center of polar plot). ΔCO lying between 30° radial sector limits, excluding data within 10% of the CO, should identify a good trending ability.

## 3. Results

We considered 50 patients for enrolment. Six patients were excluded due to an inappropriate arterial waveform signal based on an amplitude ratio and natural frequency calculation, despite the efforts to correct it; four patients were excluded because of severe mitral regurgitation (3) and tricuspidal regurgitation (1). Demographic data, origins of septic shock, and severity scores of the 40 included patients are shown in Table 1.

The mean heart rate at T1 was 83 bpm (SD 9.24) and at T2 was 81 bpm (SD 7.18). The mean MAP at T1 was 76.15 (SD 3.04) mmHg, after 15 min it was 67.6 (SD 3.01) mmHg and at T2 it was 74.5 (SD 3.34) mmHg; variations in MAP are depicted in Figure 1.

Fifteen patients received 500 mL of crystalloids between T1 and T2.

A total of 160 CO measurements were collected. The Pearson product moment correlation between PiCCO and MostCare at T1 and T2 was 0.9 and 0.83, respectively. To investigate if the variation of norepinephrine afflicted the accuracy and precision of MostCare, we performed a separate Bland-Altman analysis at T1 and T2. At T1 the mean CO was 5.38 (SD 0.60) L/min, the mean difference was 0.176 L/min, the LoA was +1.39 and −1.04 L/min, and the PE was 22.6%; at T2 the mean CO was 5.44 (SD 0.73) L/min, the mean difference was 0.053 L/min, the LoA was +1.51 and −1.40, and the PE was 27%. Agreement between cardiac output measured by MostCare (MCO) and cardiac output measured by PiCCO transpulmonary thermodilution (PCO) at the baseline (T1) and 30 min after norepinephrine reduction (T2) is shown in Figure 2.

After considering patients who received volume expansion between T1 and T2, the mean CO at T1 was 5.39 L/min (SD 0.47), the LoA was +1.09 and −0.78 L/min, and the PE was 17%; at T2 the mean CO was 5.35 L/min (SD 0.81), the LoA was +1.73 and −1.52 L/min, and the PE was 30%.

The trending ability of MostCare of overall patients is shown by data points dispersion on polar plot analysis (Figure 3): 29 data points (difference in CO between T1 and T2), were inside 10% of CO variation, indicating a substantial correspondence between the two values. Among the remaining 11 data points, the polar concordance was 82% (data between 30° angles).

## 4. Discussion

In clinical settings where hemodynamic monitoring could be useful but invasive tools are not immediately and easily available, there is a lack of knowledge about the reliability of less invasive and rapid setup methods. The purpose of our study was to evaluate the level of agreement between a calibrated transpulmonary thermodilution device (PiCCO System) and an uncalibrated one (MostCare) in detecting and tracking changes in CO measurements during “dynamic” clinical conditions related to vasopressor de-escalation in septic shock patients. MostCare uses a patented algorithm to estimate CO, but some concerns have been raised about its ability to self-calibrate in clinical situations where arterial tone rapidly changes [17].

The PiCCO system is an invasive tool which has been validated in animal models and in humans under several clinical conditions [8,29]. Its numerous advantages have already been highlighted [18]. Transpulmonary thermodilution CO assessment is based on calculations that were verified under stable setting and sinus rhythm conditions. Despite criticism of the use of PiCCO in unstable clinical conditions [30,31], many authors [8,32,33] found good agreement with the reference method, thereby also confirming its good performance in hemodynamically unstable and septic shock patients [12,13].

Notably, PiCCO needs a dedicated arterial catheter, a central venous line, and a calibration after any hemodynamic perturbation, such as vasopressor variations or fluid challenges [25]; its continuous cardiac output measurement, based on pulse contour analysis, better fits with transpulmonary thermodilution if it is re-calibrated every hour [11]. All these characteristics make this device difficult to use in clinical settings like emergency wards.

The MostCare system has been evaluated and validated in different clinical settings [23,34,35,36].

However, the literature provides only a few reports of its use in septic patients or during hemodynamic instability, with controversial results. Franchi et al. [16] showed a good correlation between MostCare and PAC on tracking changes in arterial tone by increasing and decreasing norepinephrine dosage; variations were targeted on a predefined value of arterial pressure and in selected hemodynamic stable patients. In contrast, Gopal et al. [17] showed a poor correlation with PAC in septic patients, but neither data about arterial waveform artefacts nor ultrasound heart assessments [37] were considered.

As observed by other authors [38], the change in arterial load seems to be the main determinant of differences between central and peripheral CO assessments, and this is a typical feature of septic patients. In our population, the arterial elastance and net compliance were altered by sepsis and vasopressor modifications. In this view, a hemodynamic monitoring system should be able not only to record a static value in CO, but also to track any variation in the hemodynamic setting. Some authors suggested that a less accurate measurement may be acceptable if the trend analysis is reliable [39]. In our study the norepinephrine dosage reduction was constant (0.03 mcg/Kg/min), as occurs in our clinical practice. This variation altered the equilibrium of the cardiovascular system for all the patients (Figure 1), and clinicians were free to perform a fluid challenge if needed. None of the patients needed to get a re-increase in the norepinephrine dosage. Unfortunately, the CO values at T1 and T2 were similar for the majority of our patients, thereby limiting the power of tracking analysis. Anyway, the polar plot diagram seems to confirm the trending ability of the MostCare system versus the reference method, but this has to be confirmed by larger studies. The large number of dots inside the exclusion zone represents the statistical noise due to the high level of random variability.

In our group of patients, when separately analyzed at T1 and T2, the CO values measured by the two systems showed acceptable LoA and PE outcomes. In the subgroup that received both volume expansion and norepinephrine reduction, MostCare showed a good agreement with the reference method, if compared with different monitoring systems [40]. The opportunity to get a rapid assessment and continuous monitoring of CO in a septic patient could be very useful for guiding a goal-directed approach from the early phase of presentation, in order to manage hemodynamic instability and optimize treatment. Moreover, we considered patients with septic shock originating from different sites, as the source of infection is often still unknown at presentation in emergency departments.

We only excluded patients affected by conditions that could limit the reliability of the reference method [18] and those who did not show an optimal arterial waveform due to over/under-damped signal. The quality of the arterial waveform is of paramount importance, particularly when considering the MostCare system, as it analyzes arterial waveform at 1000 points/s. Romagnoli et al. [23] found 30.7% of patients with an under-damped arterial signal in vascular and cardiac surgery. An under-damped signal can lead to a significant overestimation of the CO when measured by the MostCare system, which does not seem to be an independent operator or a ‘plug and play’ monitor. The optimization of the waveform signal to avoid resonance artifacts is time consuming and is based on mathematical calculations [24]. In our study, the system was not reliable a priori in 12% of our patients, due to arterial artifacts. In fact in 6 patients, despite the efforts to correct the quality of output waveform signal, an under-damped arterial waveform was recorded.

This study presents some caveats: the first involves the reference method. Transpulmonary thermodilution, despite extensive papers confirming its reliability, has several limitations and the differences between PCO and MCO could hypothetically be partly related to PiCCO’s performance. The second is that we performed transthoracic echocardiography rather than transesophageal echocardiography to exclude conditions that could afflict the accuracy of the reference method; transthoracic echocardiography has notably been proposed as a reference method in several studies assessing CO [34,37].

## 5. Conclusions

In septic patients, when the arterial waveform signal is accurate, MostCare and transpulmonary thermodilution systems exhibit good agreement even after the reduction of norepinephrine and changes in vascular tone or volume expansion. Thus, MostCare can be considered as a reliable and useful tool in emergency clinical settings where highly invasive devices are not promptly available.

## Figures and Tables

**Figure 1 jcm-10-00213-f001:**
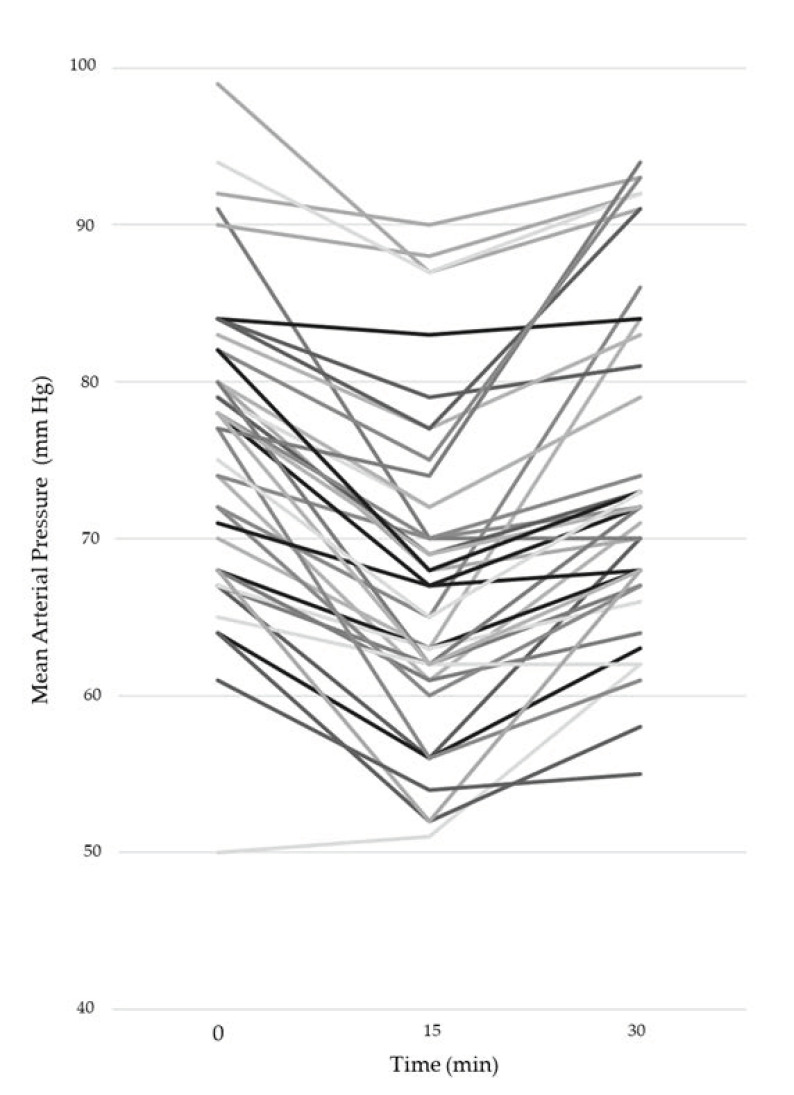
Mean Arterial Pressure changes at baseline, after 15 min and after 30 min from norepinephrine infusion reduction.

**Figure 2 jcm-10-00213-f002:**
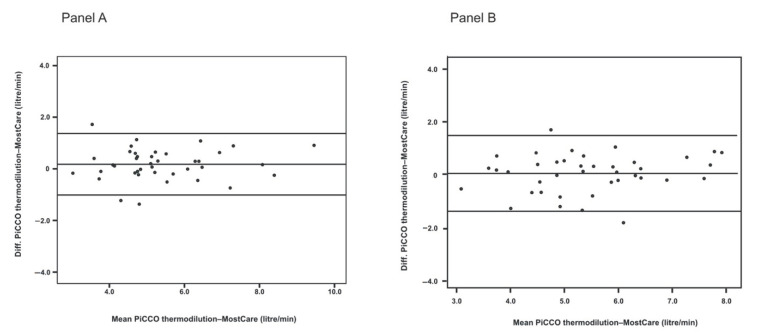
(**Panel A**) Bland-Altman plot at T1; the horizontal lines on the scatterplot represent the bias and the limits of agreement (mean bias ± 2 Standard Deviations). (**Panel B**) Bland-Altman plot at T2; the horizontal lines on the scatterplot represent the bias and the limits of agreement (mean bias ± 2 Standard Deviations).

**Figure 3 jcm-10-00213-f003:**
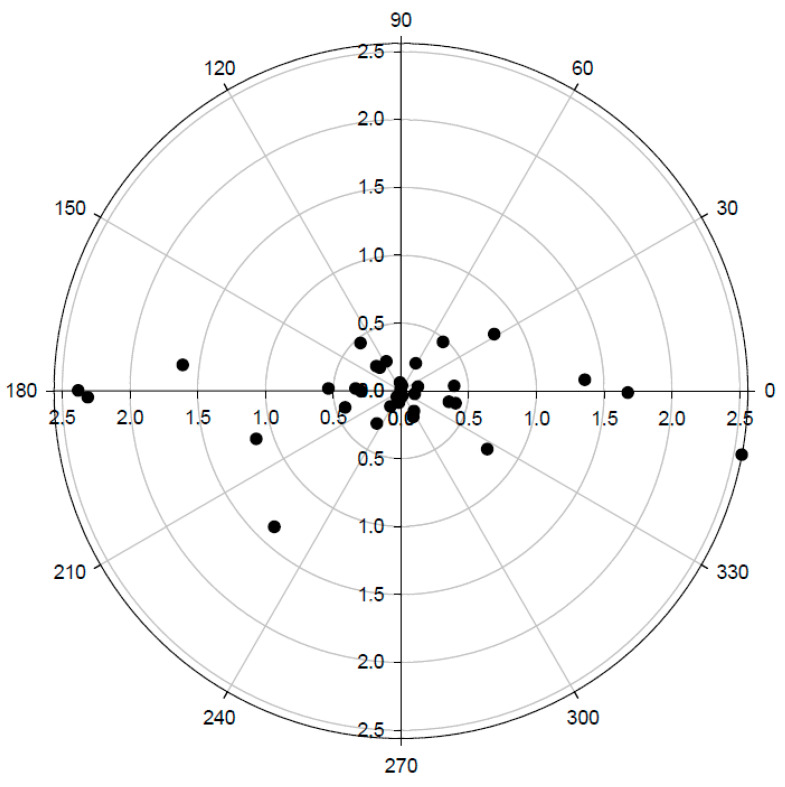
Polar plot showing the trending ability. This was obtained by converting CO variation obtained between T1 and T2 from PiCCO System (X-axis) and MstCare system (Y-axis). Radial sector limits, excluding data within 10% of CO variation, identify a good trending ability.

**Table 1 jcm-10-00213-t001:** Patients characteristics. Data are expressed as percentage and mean ± standard deviation.

Patients’ Characteristics	Values
Age (years)	64 ± 12
Gender (M/F)	19/21
Weight (Kg)	76 ± 12
Height (cm)	174 ± 15
Body surface area (m^2^)	1.83 ± 0.23
SAPS II	42 ± 9.3
Origin of septic shock	
abdominal	18 (45%)
pulmonary	10 (25%)
bacteremic	4 (10%)
soft tissues	4 (10%)
cerebral	1 (2.5%)
ob/gyn	1 (2.5%)
unknown	2 (5%)

M/F, Male/Female; SAPS II, Simplified Acute Physiological Score II.

## Data Availability

Not applicable.

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
