# Peer review of "Cardiac Output Evaluation on Septic Shock Patients: Comparison between Calibrated and Uncalibrated Devices during Vasopressor Therapy"

_jcm, 2021, doi:10.3390/jcm10020213_

Round 1

Reviewer 1 Report

This is a very difficult study to perform on patients with septic shock in the ICU on vasopressor support. The fact that you only had to exclude only 10 patients is truly amazing! I did not see a disclosure related to your involvement with the MostCare system company. Did they fund the study? If so this should be disclosed.

The study has many limitations but in some circumstances a minimally invasive cardiac output monitor may be useful even when patients are being infused norepinephrine or vasopressin. Further studies with larger numbers and the use of trans-esophageal Echo are required but this is a very tough patient population to make accurate measurements for a clinical trial on and you have done a great job on a diddicult study.

Author Response

Dear Editor,

Thank you for your revision; I'm really grateful for your encouragement.

I received unrestricted educational support in the form of payment of conferences registration fees (ISICEM, Brussels, 2015, 2016) from Vygon-Vytech, the manufacturer of the MostCare device. I do not have any financial relationship with this nor other companies. The remaining authors have no conflict of interest to declare.

The company did not fund the study and is not aware of the results.

Reviewer 2 Report

The authors in this work evaluate the value of agreement between 2 methods of measuring cardiac output and the variation of the latter during a therapeutic intervention in septic shock patients

Rather well written, the authors clearly present the objectives and methods of their study. The population studied is pertinent.

This study is quite interesting in its objectives (to show the interest of a non-invasive method of measuring cardiac output during septic shock) suffersunfortunately from many major limitations which are, moreover, well discussed by the authors themselves: the reference method PICCO is questionable, the modification of norepinephrine infusion rate does not modify cardiac output, the need for a study including a larger population. BUT  the honesty in the analysis of their work is to be underlined and in my opinion justifies the publication of this work.

Authors should specify: the frequence of fluid administration and the fluid’s volumes during measurements. Furthermore, if an echocardiogram has been performed, why not make another comparison of the agreement of the variation of cardiac output with the both device (Mostcare and Picco) vs echocradiography?

Minor remarks: please note that some references appear in the text and are not in the format of the journal.

Although this is not the question, the outcome of these patients could also be of interest.

Author Response

Dear Editor,

thank you for your revision and your useful suggestions.

Here the point-by-point response to your comments:

  • the reference method PICCO is questionable
    • Transpulmonary thermodilution, despite several papers confirming its reliability, has some limitations and we underlined these in the paper; we chose this reference method for its less invasiveness than PAC and because in the Consensus on circolatory shock and hemodynamic monitoring (2014) Cecconi et coll. equally recommended the use of PAC or transpulmonary thermodilution to determine the type of shock (Grade Level 2 and quality of evidence Low). We totally agree with you that PAC is still considered the gold standard in CO measurement.
  • the modification of norepinephrine infusion rate does not modify cardiac output
    • we followed an internal protocol that consider 0.03 mcg/Kg/min as standard dose for drug reduction. Unfortunately, despite its effect on mean arterial pressure (as depicted in fig.1), the reduction of norepinephrine did not affect significantly the CO value. If so, we could have collected more information about CO tracking ability by MostCare system.
  • the need for a study including a larger population
    • We totally agree with you; we calculated the power of the study considering that the true difference in the mean response of matched pairs is 0.55 resulting in 37 subjects to study. Maybe, if we had enrolled more patients, we could have observed an higher variability in T2 CO.
  • the frequence of fluid administration and the fluid’s volumes during measurements
    • We better clarified in materials and method section (line 122) that volume expansion was performed with 500 ml of crystalloid only once between T1 and T2. If the MAP did not raise, the norepinephrine would have been increased again and the patient excluded from the study.
  • if an echocardiogram has been performed, why not make another comparison of the agreement of the variation of cardiac output with the both device (Mostcare and Picco) vs echocardiography?
    • Thank you for your great suggestion; for the best of our knowledge, there are few papers comparing MostCare to echocardiography in CO measurement in ICU patient but they are from the same group of study and they don't consider the comparison with PICCO too. It would be of great interest to get these information from the three methods. Unfortunately the echocardiographic evaluation was performed on the admittance of patients and not during the data recording; so the conditions could have been different in the two moments. We are very thankful for your suggestion for further studies.
  • please note that some references appear in the text and are not in the format of the journal.
    • Thank you, we changed the references as requested

Round 2

Reviewer 1 Report

Revisions have improved this manuscript